# *"Africans, we know how to adapt indeed"*: Adaptations to family planning and reproductive health services in humanitarian settings in Nigeria during the COVID-19 pandemic

Emily Evens[1], Ashley Ambrose[2]*, Bamidele Bello[3], Kate Murray[1], Nadia Tefouet[2], Adesegun Fatusi[3], Bridget Nwagbara[4], Mercy Riungu[5], Tijani Maji[5], Hadiza Khamofu[4], Jean Christophe Fotso[2], Ndola Prata[2]

1 FHI 360, Durham, North Carolina, United States of America, 2 Evidence for Sustainable Human Development Systems in Africa, Yaoundé, Cameroon, 3 Academy for Health Development, Abuja, Nigeria, 4 FHI 360, Abuja, Nigeria, 5 FHI 360, Maiduguri, Nigeria

* ashleyarowe@gmail.com

**Data Availability Statement:** The datasets used for this study are available on the Harvard Dataverse and can be accessed using this link: https://

## Abstract

On March 30, 2020, the Government of Nigeria implemented its first COVID-19 related lockdown. We worked with two humanitarian projects in Nigeria, the Integrated Humanitarian Assistance to Northeast Nigeria (IHANN II) in Borno State and the United Nations High Commissioner for Refugees South-South Health and Nutrition Intervention (UNHCR-SS-HNIR) for Cameroon Refugees and vulnerable populations in Cross River State, to document the programmatic adaptations to Family Planning/Reproductive Health (FP/RH) services in response to COVID-19 and identify successes and challenges of those adaptations. A mixed methods approach including quantitative analysis of data from routine programmatic activities, qualitative data from in-depth interviews (IDIs) with project staff and process documentation of programmatic activities and modifications was used to 1) identify modifications in FP/RH services due to COVID-19, 2) understand staff perception of their utility and impact, and 3) gauge trends in key FP/RH in-service delivery indicators to assess changes prior to and after the March 2020 lockdown. Monitoring data shows notable declines in service utilization after lockdowns in antenatal care, postnatal care, and outreach campaigns, followed by a return to pre-lockdown levels by July 2020. Results show projects introduced numerous COVID-19 precaution strategies including: community sensitization; triage stations and modification of service flow in facilities; and appointment scheduling for essential services. Findings from IDIs speak to a well-coordinated and implemented COVID-19 response with project staff noting improvements in their time management and interpersonal communication skills. Lessons learned included the need to better sensitize and educate communities, maintain FP commodities and increase support provided to health workers. Deliberate adaptations in IHANN II and UNHCR-SS-HNIR projects turned challenges to opportunities, ensuring continuity of services to the most vulnerable populations.

dataverse.harvard.edu/privateurl.xhtml?token=
66a1a631-61df-4349-9856-02a90b134a90.

**Funding:** This research was funded under the Grant funded by Family Health International (FHI) under Cooperative Agreement/Grant no 7200AA19CA00041 funded by United States Agency for International Development (USAID). The content of this paper does not necessarily reflect the views, analysis of policies of FHI 360 or USAID, nor does any mention of trade names, commercial products, or organizations imply endorsement by FHI 360 or USAID. The funders had no role in study design, data collection and analysis, decision to publish, or preparation of the manuscript.

**Competing interests:** The authors have declared that no competing interests exist.

## Introduction

Provision of family planning and reproductive health (FP/RH) services in humanitarian settings is a complex but necessary endeavor [1]. A humanitarian setting is one in which an event or series of events has resulted in a critical threat to the health, safety, security and well-being of a community or other large group of people [2]. In these settings, access to RH services often decreases [3] while RH needs increase [4]. Women and girls in humanitarian settings are particularly vulnerable with many losing livelihoods, education opportunities, and family, social and structural support systems, placing them at increased risk of unwanted pregnancy, unsafe abortion, sexually transmitted infections (STIs) including HIV, and maternal illness and death [5]. Comprehensive FP/RH services, including FP services, antenatal (ANC) and postnatal care (PNC), post-abortion care, STI testing and treatment, and gender-based violence (GBV) prevention and response, are critical to meeting the health needs and rights of women and girls in humanitarian areas.

The COVID-19 pandemic exacerbated the FP/RH needs of women and girls in humanitarian settings. FP/RH services in humanitarian areas suffered significant disruptions given their modes of delivery which often include community meetings, outreach activities, and peer support groups [6]. The COVID-19 pandemic has been linked to decreased access to and use of sexual and reproductive health services among women and girls [7] largely due to disruptions in supplies of contraceptives [8] and a limited health workforce and resources [9]. Fear of contracting COVID-19 in health facilities could further limit access to services [10].

The first confirmed case of COVID-19 in Nigeria was reported on February 27, 2020. As of June 13th, 2022 there had been 256,352 confirmed cases and 3144 confirmed deaths [11]. In March 2020, the Presidential Taskforce for COVID-19 was commissioned, travel bans were implemented for some states, and some states restricted the size of gatherings [12]. A national lockdown was ordered on March 30, 2020, with additional states (including Borno and Cross River states) declaring lockdowns during April 2020 [13]. An initial lockdown period of two weeks was then extended by an additional three weeks [14]. In Borno and Cross River states, restrictions included closure of borders to avoid movement in/out of the state, closure of schools and churches, limiting the number of passengers on public transportation, limiting gatherings to five people, and closure of government offices with exceptions for essential services like hospitals and law enforcement. From February to early May 2020, Nigeria saw only 2950 cases with a positivity rate of 1/43 per 100,000 people and few confirmed cases in humanitarian settings in the country's northern region [15]. But this quickly changed and by July 2021 Nigeria was the 4th most affected country in Africa with a case fatality rate of 1.3% [16].

The on-going insurgency in North-East Nigeria has lasted for more than a decade and has put up an estimated 8.7 million people in need of humanitarian aid [17]. Additionally, conflict between armed separatist and government forces which began in Cameroon in 2017 has led to a rapid influx of Cameroon refugees in the South-South, North-Central and North-East Nigeria with up to 77, 000 refugees currently residing in Nigeria [18]. In the Northeast states affected by the Cameroon refugee situation, especially Borno and Cross River, Taraba and Benue states, COVID-19 is a crisis within a crisis. In these areas the pandemic has further strained an already weakened health system deeply affected by humanitarian needs and insecurity. Stay-at-home orders, border closures, and other movement restrictions, along with the preexisting situation of poor health seeking behavior, stigma, and low socio-economic status created a fragile economic situation and delays in accessing healthcare.

To date only a few studies [10, 19] document the extent to which travel and gathering restrictions and service shut-downs related to COVID-19 affected FP/RH services in the country. Adelekan et al. assessed the provision of reproductive, maternal, and child health (RMNCH) services before, during and after COVID-19 lockdowns finding both a modest

reduction in service delivery as well as stockouts, transportation challenges and harassment of health providers by law enforcement agents [10]. Balogun et. at found that despite challenges such as drug and contraceptive stock-outs, limited personal protective equipment and transportation difficulties, the large proportion of primary health centers that continued to provide RMNCH services demonstrated resilience [20]. However, data specific to RMNCH care in humanitarian settings in Nigeria, during COVID are limited.

The purpose of this paper is to 1) identify modifications in FP/RH services in two humanitarian settings in Nigeria during the first year of the COVID-19 pandemic, 2) understand staff perceptions of their utility and impact, and 3) assess trends in key FP/RH service indicators during this period. Policies related to FP/RH service provision during this time period in Nigeria were also used to contextualize results.

## Project background

Two projects supporting FP/RH services for women in humanitarian settings in Nigeria in the time of COVID-19 were part of this study. The IHANN II (Integrated Humanitarian Assistance to Northeast Nigeria, April 2019 –December 2020) project is a multisectoral humanitarian response in Borno State that provides FP, RH and maternal, newborn and child health services to conflict-affected and displaced Nigerians. IHANN II works primarily in camps for internally displaced people (IDP) but also targets host communities with interventions. IHANN II health facilities are located within highly populated camps for internally displaced persons in hard-to-reach areas of Borno State. Due to the prolonged crisis in Northeast Nigeria, these are mostly supported by NGOs. The UNHCR-SS-HNIR (United Nations High Commissioner for Refugees South-South Health and Nutrition Intervention, April 2019 – December 2020) project operated in Cross River State and provided essential services including FP, ANC and PNC to Cameroonian refugees living in settlements and host communities. (UNHCR SS-HNIR also operated in Benue State, but that site was not included in this study. The activity ended in December 2020). Health facilities under the UNHCR-SS-HNIR project consist of public and government-run health facilities in both host facilities and camps. Table 1 below summarizes the settings and FP/RH services provided by each project.

**Table 1. Summary of the study settings and services they provide.**

|  | IHANN II | UNHCR SS-HNIR |
|---|---|---|
| Population served | Internally displaced persons living in camps and host communities | Cameroonian refugees living in settlements and host communities |
| State in Nigeria | Borno | Cross River |
| *Health facilities* | **NGO-run health facilities** | **Public health facilities** |
| **FP/RH services provided** |  |  |
| Family planning* | X | X |
| Antenatal care | X | X |
| Post-natal care | X | X |
| STI | X |  |
| GBV | X |  |
| Post-abortion care | X |  |
| Obstetric care | X | X |

*FP methods offered through direct service provision include: condoms, oral contraceptive pills, injectables and implants. Referrals were made for IUCD, and surgical methods such as bilateral tubal ligation and vasectomy. Additionally, referrals may be made for FP methods that were not available at the time of services due to stockouts.

## Data and methods

We conducted a mixed methods study to identify adaptations made to FP/RH services during the COVID-19 pandemic and quantify the impact of COVID-19 on key FP/RH indicators. Data were collected through: 1) Review of policy and recommendation documents, 2) Analysis of trends in FP/RH data routinely collected service statistics by projects, from January 2020 (two months prior to lockdown in Nigeria) through December 2020 and 3) Key informant interviews with project staff including: clinical health care providers, programmatic staff and outreach workers. (See Table 2 for the distribution of key informants by project and position type).

Policies and recommendations which guided provision of FP/RH services in humanitarian settings during the initial months of the COVID-19 pandemic were identified by project staff. The policy and recommendation documents were then reviewed by the study team and recommendations from each document were extracted using an analysis matrix. Recommendations were then synthesized across the documents and summarized.

The study team analyzed routine programmatic monitoring data from the two projects to describe monthly trends in the provision of FP/RH services during COVID-19 from January 2020 through December 2020. IHANN II data represents nine (9) primary health facilities across four Local Government areas (LGA)s in Borno State and UNHCR-SS-HNIR-SS-HNIR data represents ten (10) primary health facilities across six LGAs in Cross River state. Monthly numbers of clients for each of these services were collected for each facility: ANC attendees, deliveries attended by skilled personnel, women receiving FP services, individuals reached with RH community outreach, and pregnant women who received HIV counselling and testing. Service utilization data was collated from project registers as a part of projects' routine monitoring system. Data collected in health facilities is then validated by project monitoring and evaluation staff during field visits and sent to project offices for compilation. Data was then cleaned to identify any outliers or inaccuracies using Microsoft Excel. Analysis was conducted using Excel to identify changes by month and compare two months prior to March 2020 when COVID was declared a global pandemic by the WHO and restrictions and closures were instituted in Nigeria. All analyses were descriptive using frequencies, means and descriptive trend analyses through graphical presentation. All data presented have been verified.

Research assistants conducted 40 semi-structured in-depth interviews to gather information on the effect of COVID-19 on FP/RH service provision, project modifications, staff recommendations and lessons learned. Twenty staff were purposively selected from each project and stratified to ensure representation of clinical/health care providers, project staff and outreach workers. The phone-based, semi-structured interviews, were conducted in English by a local trained data collection team, and were audio-recorded, transcribed, and electronically stored with protective passwords. All responses were analyzed with an analysis matrix using thematic approach involving the identification, examination, and interpretation of project adaptations, feasibility of interventions and recommendations for the future. Inter-coder reliability was obtained by having the research team members enter multiple transcripts into the

**Table 2. Key informants by project and position type.**

|  | IHANN II | UNHCR SS-HNIR |
|---|---|---|
| **Healthcare provider** | 6 | 6 |
| **Program staff** | 6 | 7 |
| **Outreach worker** | 7 | 8 |
| **Total** | 19 | 21 |

matrix together until they came to agreement as to how codes would be used, then periodically reviewing coding, and adjusting as needed.

### Ethical statement

Ethical approval for this study was granted by FHI 360's Office of International Research Ethics as well as by Nigerian State level Ethical Review Boards in Borno and Cross River states. All quantitative data obtained from project sites is facility-level and de-identified. Key informant interviews were conducted with adult project staff and informed consent was obtained for each interview.

## Results

### Service delivery policies for humanitarian settings during COVID-19 pandemic

Project staff identified eight policies/guidelines which were available to them during the study period and informed service delivery at that time. These documents provided guidance on how services should operate in the context of COVID-19. The majority (five) of the documents [21–25] were international guidance from sources such as the World Health Organization and international non-governmental organizations. The international guidance included guidance from Jhpiego, "Ensuring Quality Family Planning Services during COVID-19 Pandemic" [21]; from the World Health Organization, "COVID-19: operational guidance for maintaining essential health services during an outbreak: interim guide" [22], "Safe Ramadan practices in the context of COVID-19: Interim guidance" [23], and "Continuing essential Reproductive, Maternal, Neonatal, Child and Adolescent Health services during COVID-19 pandemic" [24]; and from Pathfinder International, "Technical Guidance: Family Planning during COVID-19" [25]. Three national or state policies for Nigeria [26–28] issued in March and April 2020 also informed health service adaptations. These included guidance from Health Sector Nigeria, "COVID-19: Preparedness and response priorities for the ongoing humanitarian situation in Northeast Nigeria" [26]; from the Gender-Based Violence (GBV) Sub-Sector Nigeria, "Guidance Note: GBV Service Provision in the Context of COVID-19" [27]; and from Borno State Ministry of Health, "COVID-19 Preparedness and Response Plan" [28]. Only one document addressed humanitarian settings.

Both international and national/local guidance provided recommendations for health service delivery. Across all policies, there was an emphasis on maintaining access to and continuity of essential FP/RH services including antenatal and postnatal care, childhood immunizations, GBV services and FP. Several policies suggested use of telehealth/telemedicine when possible. Guidance on infection prevention and control (IPC) to protect client and community health as well as health care providers stressed the importance of personal protective equipment (PPE) such as facemasks. Recommended changes to service delivery to decrease risk of COVID-19 transmission included social distancing, triaging patients, and reducing the number of patients waiting at the health facility at one time (for example, through appointments). Policies also emphasized the importance of maintaining availability of contraceptives and preparing for supply chain disruptions. Recommendations included using multi-month dispensing, advance provision of emergency contraception, and/or shifting to self-administered or longer acting methods.

In addition, both IHANN II and UNHCR-SS-HNIR were guided by a three-level categorization of health care for planning purposes developed by the IHANN II project. Category 1 included regular day-to-day health care services for common diseases like malaria, eye and ear

infections, and diarrhea. In response to COVID-19, facilities implementing these services were advised to enforce social distancing to reduce the number of individuals in clinics, and the use of PPE and provide door-to-door health care services within camps. Category 2 included routine scheduled health care services including ANC, FP, and previously diagnosed non-communicable disease management. FP services continued to be offered but only by appointment to reduce crowding within health facilities. Category 3 patients seeking emergency health care services were continuously seen in the health facility during COVID-19 with the use of extensive PPE reserved for extended contact with patients during care.

### Impact on FP/RH service delivery

IHANN II data shows substantial declines in ANC service utilization following the April 2020 [29] lockdown, with a continued decline until June 2020 and a return to pre-COVID-19 levels in July (Fig 1). HIV counseling and testing also had declines in April and May before bouncing back in June. Provision of FP services and deliveries attended to by skilled personnel showed little effect.

The UNHCR-SS-HNIR project data is more variable with the most notable effects being a substantial decline in HIV counseling and testing and smaller declines in ANC and FP following the March 2020 lockdown; all of which returned to pre-lockdown levels by June 2020 (Fig 2). Deliveries by skilled personnel showed very little change.

Both projects experienced substantial declines in individuals reached through outreach services during lockdown, hitting a low point in the month of May, but these numbers similarly increased in the months following lockdown (Fig 3 and 4). For example, within the UNHCR-SS-HNIR project, the total number of community members reached through

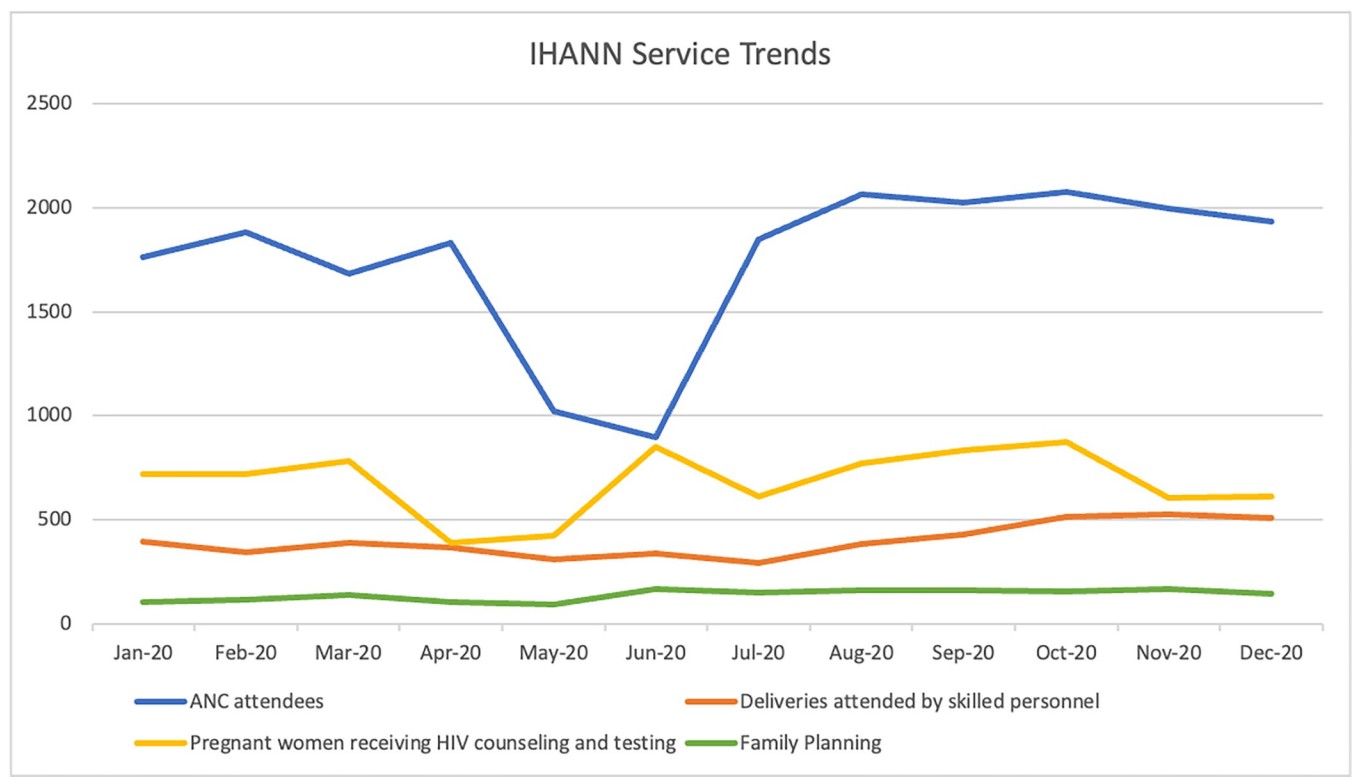

**Fig 1. IHANN II FP/RH service trends.**

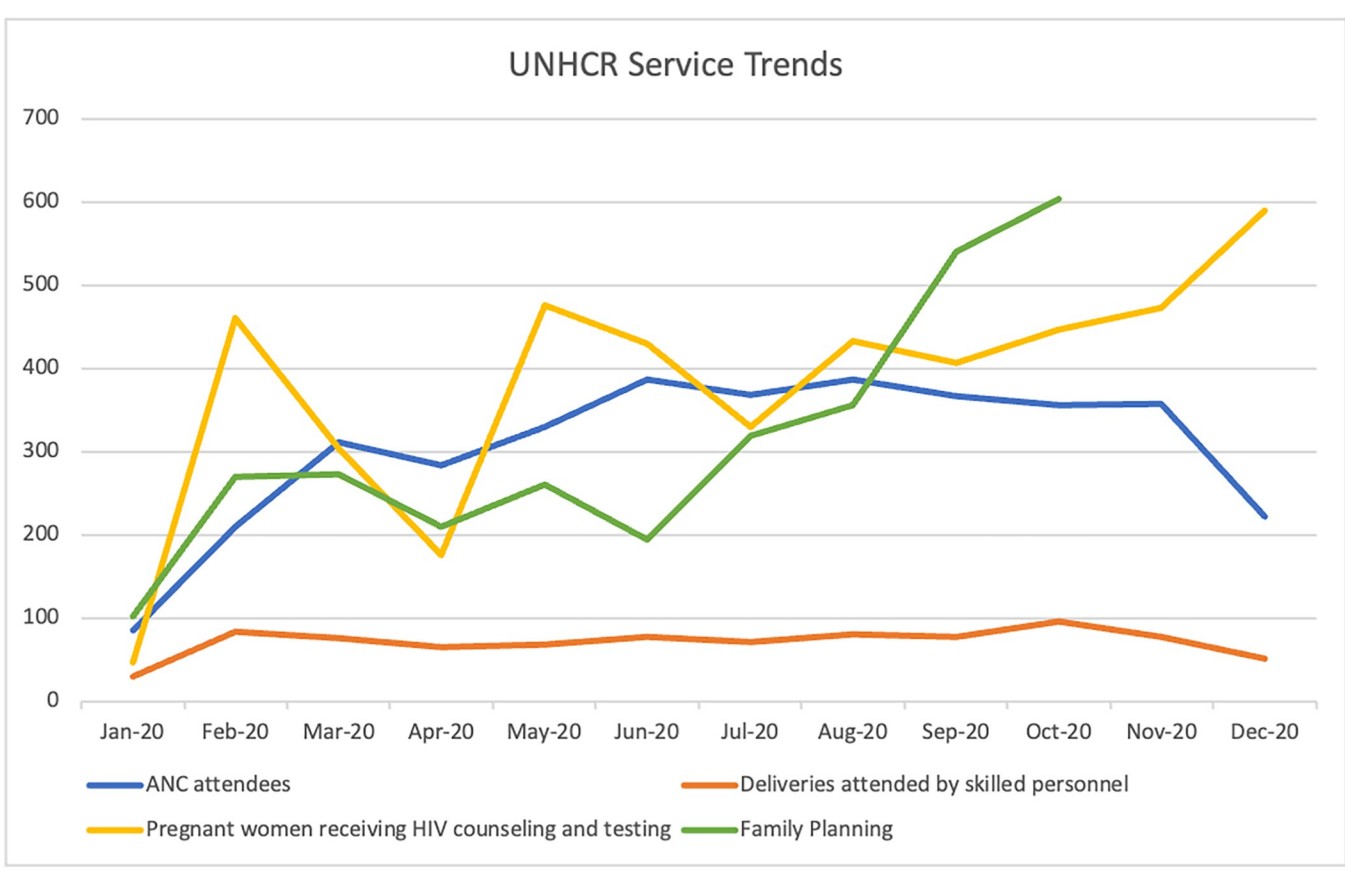

**Fig 2. UNHCR-SS-HNIR FP/RH service trends.** *UNHCR-SS-HNIR family planning data for November and December 2020 not available.

community outreach services went from 225 in April to 0 in May during the initial lockdown, to 702 in June once movement resumed.

## COVID-19 adaptations for facility and community-based activities

Data from qualitative interviews provides details on the adaptations implemented, the attitudes of project staff towards them and additional recommendations and lessons learned.

**Infection prevention and control adaptations.** Both projects implemented COVID-19 prevention and control measures at the facility level, including the creation of a triage system which filtered patients with COVID-19 symptoms and sent them to specific clinicians for consultation to reduce contact with others. Additionally, waiting areas where triage was conducted was open to allow for ventilation and seating was organized to observe social distancing. Additional infection control measures included handwashing stations, enforcing social distancing and mask-wearing, and the use of PPE among health workers. At IHANN II facilities, interview respondents also noted the use of chlorinated water, temperature taking, implementation of a COVID-19 screening questionnaire at triage, and isolation of individuals with COVID-19 symptoms. Key informants at UNHCR-SS-HNIR facilities also mentioned weekly review of triage data and printing and posting COVID-19 protocols in the health facilities.

Key informants also described COVID-19 prevention and control measures implemented at the community level. In both project communities, sensitization campaigns educated community members on the importance of handwashing, mask wearing and social distancing. In

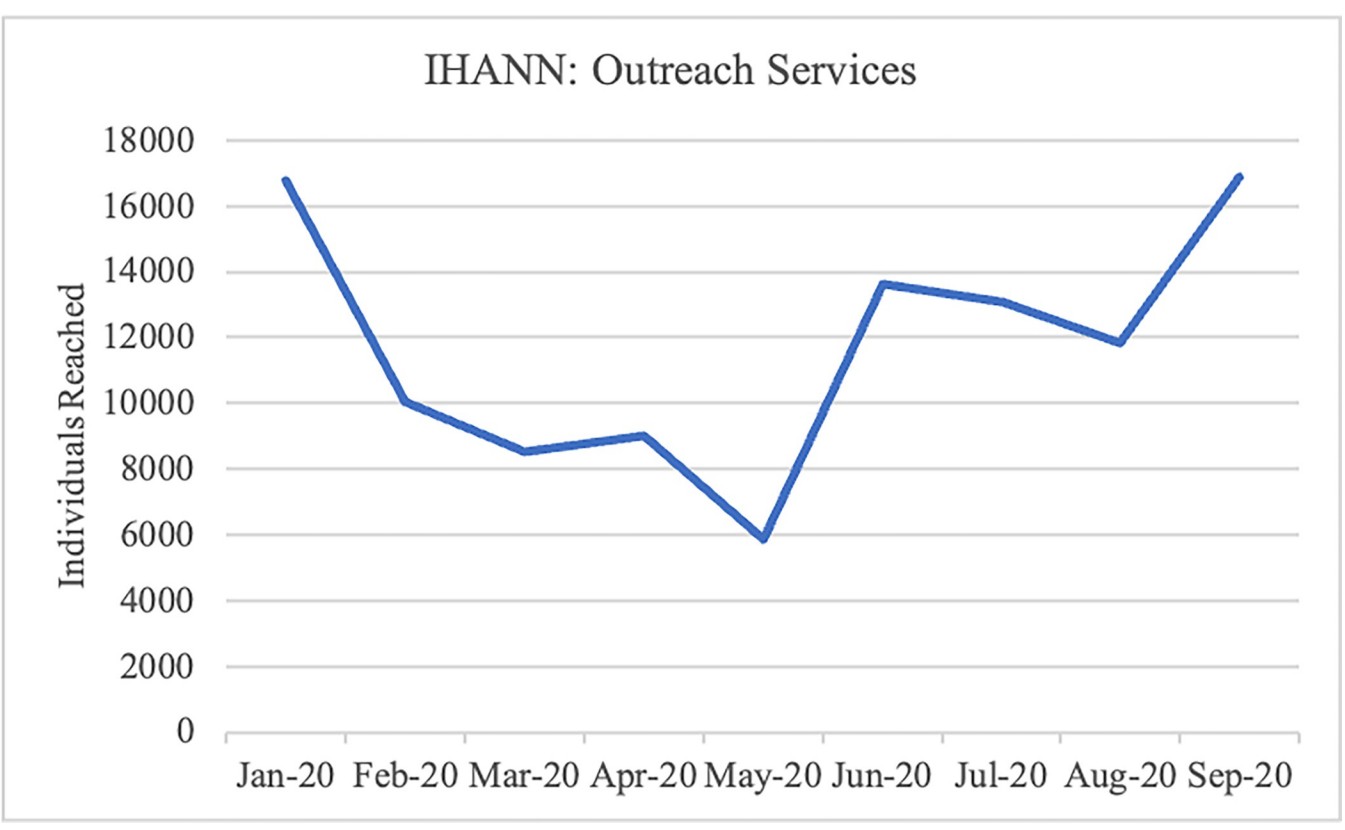

**Fig 3. Individuals reached in outreach services: IHANN II.**

UNHCR-SS-HNIR communities, handwashing stations with potable water were set up in strategic locations, such as community halls, the chief's palace and businesses. Face masks and hand sanitizer were also manufactured locally. One member of the UNHCR-SS-HNIR program staff states:

> *"We call it awareness creation, so it helped people, you know, everybody trying to get aware, support your neighbor to make sure everybody understands, if you cough, if you sneeze, if you watch people sneeze, if you observe, you just report and all that. So, . . . it gave us a very big hmmm opportunity to create more awareness and increase uptake.*

**Workflow adaptations.** Both projects made extensive modifications to the flow of health service provision. Prior to COVID IHANN II staff were based in the field sites continuously for six weeks sharing accommodations, kitchen and dining areas and washrooms, followed by five days off duty outside of the camps. During COVID -19 they were divided into small groups who worked two-week rotating shifts in the camps to reduce the number of staff in facilities at once and promote social distancing. IHANN II also made additional modifications to client flow such as batching clients into groups that were allowed to wait and those who were to return to facilities for services later, modifying client visits by providing appointments for ANC and FP services, increasing the number of ANC and PNC clinic days, and providing multi-month dispensing of medications and FP methods to reduce frequency of visits to facilities. The UNHCR-SS-HNIR project adapted workflows by rotating staff or having them work remotely to reduce the number physically present at once. Additionally, the

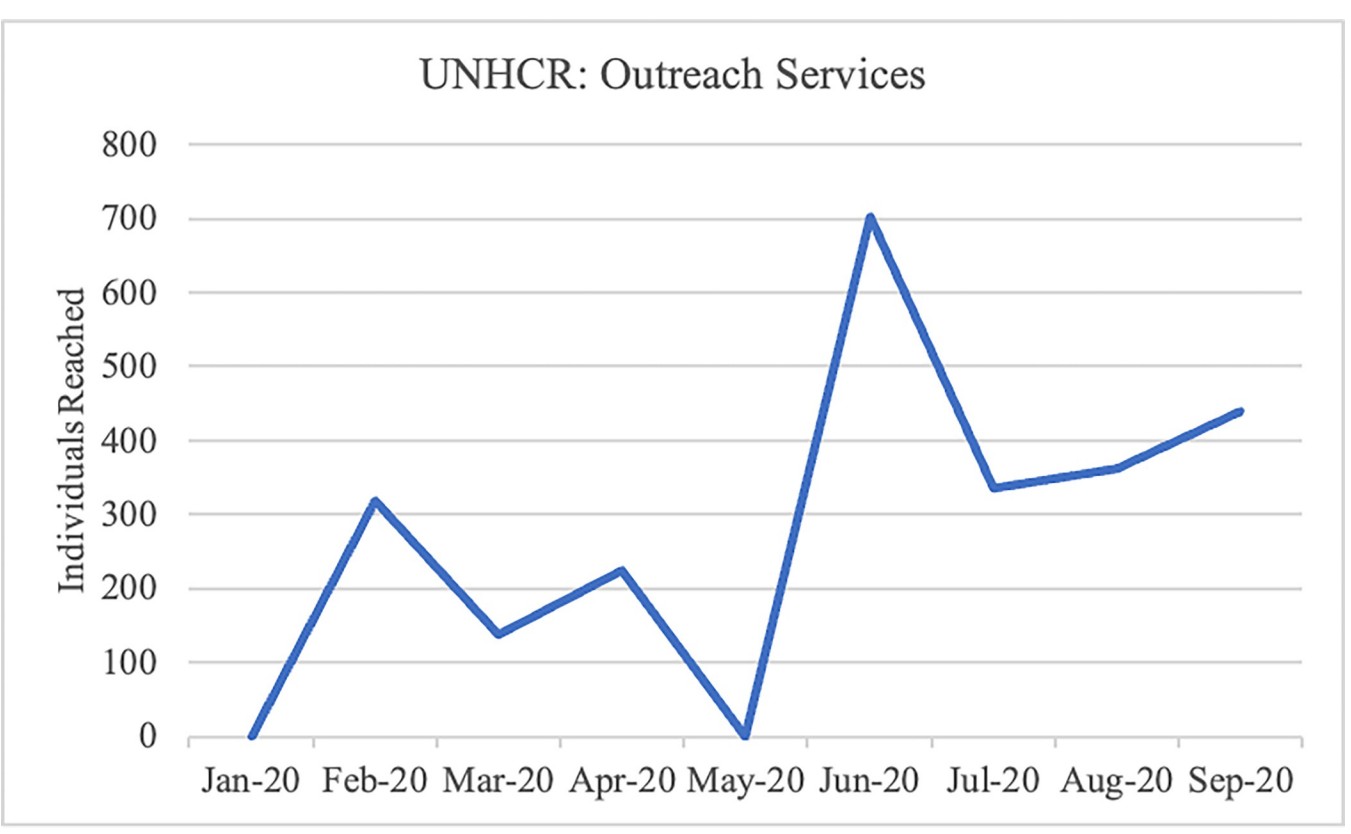

**Fig 4. Individuals reached in outreach services: UNHCR-SS-HNIR.**

UNHCR-SS-HNIR modified how FP/RH services were provided such as: replacing group counseling with individual sessions, providing tailored health education in small groups and door-to-door, conducting home visits instead of facility-based services to both discuss COVID-19 and RH/FP, providing clients with appointments when they came to facilities but were not able to be seen, providing ANC every day rather than only on selected days and times, conducting outreaches to promote the injectable contraceptive method *subcutaneous depot medroxyprogesterone acetate* (DMPA-SC), using the phone instead of in-person activities and switching programmatic staff to teleworking when possible.

Participants from both projects noted the modifications to the flow of health services worked well overall and noted that projects and staff were creative in coming up with solutions. An outreach worker from the UNHCR-SS-HNIR project states:

*"it's very, very important that as an implementer or as a programmer, if. . . there is an emergency. . .any situation, [we] find out how you can. . . come up with strategies that will help you also to address that situation on ground."*

Furthermore, a project staff member from UNHCR-SS-HNIR described how these workflow adaptations addressed project needs:

*"So even. . .at times, when we had to stay at home, the system was always on to give us information that, ok there is need for this here, there is need for this here and we work towards ensuring that those needs were made available."*

**Providers' roles.**   COVID-19 impacted the work of facility-based service providers in many ways. Both IHANN II and UNHCR-SS-HNIR staff noted they took on additional responsibilities including providing COVID-19 education and sensitization, implementing, and enforcing COVID-19 prevention measures, ordering and tracking PPE and supporting or coordinating local COVID-19 response. Across both projects, triaging clients was typically described as *"tedious", "complicated"* or *"difficult"*. One IHANN II health provider states:

*"[The] triage station was making the work more complicated, the fact that you have to watch yourself well, watch the people you are attending, you still needed to create that awareness."*

An IHANN II program staff member adds:

*"There were a lot of procedures that sometime when I look at it, I say wooo, this is a little bit tedious and that actually slows the entire process because this is something that was taken seriously into consideration, we don't want to keep any gap"*

IHANN II and UNHCR-SS-HNIR staff also cited staffing shortages due to health care providers working on rotation in order to observe social distancing guidelines or due to changes in transport. These staff shortages, alongside additional prevention measures like reducing the numbers of patients who were allowed to wait, social distancing, and the need to see patients individually, sometimes resulted in increased waiting times for patients. A few participants cited negative changes to the quality of care due to longer waiting times, reduction in the numbers of staff in facility and community activities and FP/RH clients receiving less attention as the priority was given to COVID-19. One UNHCR-SS-HNIR outreach worker states:

*"Though they will come to the facility, but the care and the attention they needed to get from us, you know some after waiting, after waiting outside making a queue, washing their hands and all of that some may be tired and they will end up leaving, so it really affected our work, the inflow of patients actually reduced and then the care and the attention that the people are supposed to get at that particular point also reduced because all attention and priority was just given to maybe Covid-19 protocol and all of that."*

Despite these challenges, most participants said these new responsibilities did not affect their normal FP/RH work with a few participants noting they enjoyed the new work and reorganized their schedule to make time for new responsibilities. One program staff member from IHANN II states:

*"COVID did not really come like a burden, it just came like something to implement, something to integrate"*

Additionally, about half of all IHANN II participants and the majority of UNHCR-SS-HNIR respondents noted that once adaptations were made, services were sufficient to meet the needs of the beneficiaries due to adequate human resources and commodities following lockdown and their ability to quickly address challenges.

**Outreach services.**   According to several IHANN II and UNHCR-SS-HNIR staff, the number of community outreaches were initially limited due to the need to adhere to COVID-19 prevention measures such as lockdowns or travel restrictions, increases in the cost of transport, and shortages in medicines. The majority of those from UNHCR-SS-HNIR noted their

work was affected in various ways by COVID-19. Key informants noted that some community outreach and project activities were stopped when staff were not able to travel to the field due to lack of identity cards required for health care workers, and when the office was shut down due to heavy COVID-19 burden among staff. IHANN II staff noted that community health campaigns were implemented door-to-door instead of through large group events. Despite these changes, nearly half of IHANN II staff, mostly outreach workers, said that COVID-19 had no impact on their responsibilities.

**FP commodities.** According to key informants, both IHANN II and the UNHCR-SS-HNIR projects experienced disruptions in the FP supply chain due to challenges with transport such as lockdown and bans on interstate movement. According to KII's these disruptions caused some patients receiving care under the IHANN II project to change methods as contraceptive implants were not always available. The UNHCR-SS-HNIR project experienced minimal stockouts as last-mile delivery of contraceptive commodities was provided by a development partner. Despite initial disruptions, however, the projects were largely able to quickly restock, leveraging relationships with other partners, the government, and the UN. As one IHANN II health care provider stated, *"We had our supplies, and we still have them."*

## Opportunities and challenges

Participants from both the IHANN II and UNHCR-SS-HNIR projects noted that COVID-19 turned out to be an important opportunity to improve organizational processes and grow professionally. Staff from both projects reported they learned more about infection prevention and control. IHANN II participants stressed how the importance of handwashing was emphasized. One IHANN II health care provider states:

> *"...with the [frequent] washing, you realize that 'mehn' your hand really gets dirty because, at no point [when] you wash your hand, you wouldn't find a lot [of] dirt, yea. So, that's it, this hand hygiene seriously, my attitude and then thoughts towards hand hygiene really changed."*

Due to changing responsibilities during the pandemic, IHANN II and UNHCR-SS-HNIR participants noted that they learned concrete new skills like triaging. UNHCR-SS-HNIR participants also reported learning how to collect samples, how to make hand sanitizer and clean equipment with bleach, and how to tell the difference between suspected, probable, and confirmed COVID-19 cases. IHANN II participants reported that they learned the importance of time management and the importance of adaptability to readjust strategies when faced with a challenge. Program staff members state:

> *"So, the pressure was really high but then with time, you know, Africans we know how to adapt indeed."*

> *"We must continue to explore every day of our work time in the humanitarian [setting] because nothing, nothing is constant. We should just continue [to] work well, and in the case of any challenges or any pandemic we will readjust our strategies."*

Participants from both projects also reported they strengthened their interpersonal relationships and communication skills and gained experience working with unique patient populations, overcoming communication barriers, managing people, and educating and sensitizing patients in the community. UNHCR-SS-HNIR participants used COVID-19 awareness talks as an opportunity to promote health-seeking behavior. Some participants noted that ANC visits and facility-based delivery increased during COVID-19 because community-based

activities conducted to communicate health risks and engage communities related to COVID-19 were also an opportunity for health care workers to refer pregnant women to facility-based delivery care. Another UNHCR-SS-HNIR program staff member noted that engaging the community allowed people to take collective responsibility for their health:

> "We use the awareness creation we are doing. . .because of COVID, at the host community level, we use the same opportunity to talk to the elderly to go for [a] medical check, you know, checking of their BP [blood pressure], sugar and other things. So, we use that avenue to cover a lot of health areas which ordinarily you know, we weren't capturing, do you understand? so that if you are sick, you must go to the facility and all these things must be checked and if you observe that is feeling this way or having this kind of symptoms you first encourage the person to use the facility."

In addition to improvements in employee capacity and skill, projects developed new processes that will have long-lasting benefits beyond COVID-19. One IHANN II health care provider noted that the triage system allowed providers to not rush and give their *"maximum for every patient"*. They state:

> "And at the same time, I now spend more hours attending to patients so, in as much as the waiting time of patient increased, the time spent attending to them has also increased yes, in between. . . each patient we have to disinfect. Yes, you have to disinfect the consultation table and all that so. . .it's quite cumbersome, yes, yes it's quite cumbersome but we are, we've been very much up the task".

## Challenges

Participants from both projects expressed how inadequate investment in FP/RH services that existed before the pandemic continued to serve as a stressor throughout the pandemic. Human resources, adequate health facilities, materials and commodities for FP/RH must be prioritized.

Inadequate support for people working in humanitarian settings also continues to be a challenge, particularly in the wake of COVID-19. As human resources are stressed, it was recommended that volunteers and other staff be provided with greater opportunity for advancement and growth. UNHCR-SS-HNIR and IHANN II staff noted the importance of providing regular training on COVID-19 and increasing allowances/salaries for essential workers such as community outreach and health care workers. An IHANN II staff member states:

> "It will also be good to increase remuneration. Yes, for healthcare workers, who have been at the forefront of this. . .fight, because for one in humanitarian crisis setting, the major challenge is their insecurity, security threat abounds. So now in addition to that, we have COVID-19 to contend with too. So, it's double trouble for each and everyone us."

Investment in the health and wellbeing of project staff was noted as important. IHANN II staff suggested providing psychological support for staff given fear of COVID-19. Prioritizing the provision of COVID-19 vaccines for staff was also recommended.

Participants from both projects also noted the need for increased education and sensitization in the community in relation to both COVID-19 and FP. Additionally, staff from both projects also noted the importance of activities to strengthen FP/RH services, aside from COVID-19 related issues including: engaging both men and women in community

conversations about FP/RH services and partnering with political and religious leaders and town criers to engage community members. Stressing the influence of community outreach, participants also encouraged increased investment in community outreach activities. One UNHCR-SS-HNIR program staff member describes their efforts to engage men around family size and family planning.

*"Now when we observed that the men were like you know, [women] need to hmmm, give birth to more children so that our family too will be big. We have lost so many of our brothers and sisters, so our famil[ies] are getting smaller and all that. But when we started bringing them in to let them know that ha, this woman cannot just, you don't have enough to eat, the CDI [cash based intervention] is barely nothing. You are not doing something that will even take care of you, your children, your wife and then you want to get more children? So, we tried to allow them to access their economic power, by so doing, they were able to support and give their consent to their women to go and access the FP for themselves."*

## Discussion

FP/RH is a key, and often unfulfilled need of women and girls in humanitarian settings [30, 31]. Prior to COVID-19 there was a large unmet need for FP/RH services among internally displaced women and girls in Nigeria [32–34]. This is especially true for adolescent girls and young women who are more vulnerable to poor sexual and reproductive health outcomes and have few services tailored to their specific needs [33, 35, 36]. The COVID-19 pandemic further exacerbated this pre-existing need, putting additional strain on an already fragile health system [37]. There is little data related to FP/RH service provision in humanitarian settings during COVID-19 in the literature [38], with one study finding decreased utilization of RH services during the initial six months; while longer-term data is not available [37]. Despite little evidence from the COVID-19 pandemic, studies show similar trends in utilization of FP/RH services during other health emergencies, including the West African Ebola epidemic where, despite an initial decrease in services, health systems were largely able to maintain consistent utilization of RH services compared to previous levels [39].

Negative outcomes of COVID-19 on FP/RH service provision were largely spared in our study's two project areas, despite challenging circumstances. National and state level policies were issued in March and April 2020, providing guidance and priorities on continuing health service provision, and both the IHANN II and UNHCR-SS-HNIR projects were able to implement most of the existing policy recommendations. Programmatic monitoring data show a decline in the number of clients seen monthly corresponding to implementation of lockdowns with a subsequent rebound in both projects and all examined services to levels at or greater than pre-COVID levels following the relaxation of lockdown restrictions. Qualitative data shows that adaptations to service delivery enabled services to be provided in new ways that addressed client and provider safety.

In the IHANN II project, the provision of ANC care was the most affected service, with a notable decline in April that reversed starting in June. Reasons for this are unclear, it is possible that clients did not view this service as essential, it is also possible that the project had challenges providing ANC care early in the COVID-19 timeline that improved later as modifications to service delivery modalities were put in place. HIV testing and counseling follow this same pattern but less dramatically. FP provision, while relatively low compared to other services, holds steady throughout this period. Under the UNHCR-SS-HNIR project, HIV counseling and testing shows the largest decline from 303 women counseled and tested for HIV in March to 176 in April. Counseling and testing began to increase again to 476 in

May as both projects received support from the Global Fund and PEPFAR partners providing HIV care and treatment to integrate ANC services with HIV testing for pregnant women. ANC and FP provision show smaller dips. Both IHANN II and UNHCR-SS-HNIR experienced their lowest provision of outreach services in May, with dramatic increases in June following the relaxation of lockdowns. In both projects deliveries attended by skilled personnel were the least affected service. While the nature of our data collection does not enable the determination of causality, this seems to point to projects' ability to ensure availability of commodities, supplies and providers, as well as clients who trusted the health care system.

Results from qualitative interviews speak to an overall well-coordinated and implemented COVID-19 response that was able to rapidly address the challenges of lockdowns at multiple levels. Project staff describe adaptations that increased access to key FP and RH services by providing them in new venues, at new times, and in ways that enhanced patient access and control. Specifically, adaptations to service delivery at the community level such as shifting from community or group counselling to door-to-door and individual activities and tailored health talks allowed activities to continue in a format that was safer for facility- and community-based project staff and beneficiaries. Facility level adaptations such as multi-month dispensing, changes in client flow such as offering appointments and increasing the number of service days ensured provision of key FP/RH services.

While data from both projects described how adaptations were put in place to ensure the continuation of services, more participants from IHANN II noted they were unable to meet the needs of their beneficiaries compared to UNHCR-SS-HNIR. IHANN II operates primarily NGO-supported health facilities in highly populated camps for internally displaced persons in hard-to-reach areas within an extremely fragile security situation while UNHCR-SS-HNIR is implemented within public, government-run health facilities in host communities and camps under minimal security constraints. It seems plausible, therefore, that contextual differences resulted in different service delivery trends between the two settings. However, it should be noted that staff from both projects described that modifications were overall effective, well-coordinated and implemented in a timely manner.

Our results are in alignment with other studies that link flexibility and innovation with the ability to ensure continuity of FP/RH services in humanitarian settings [40] including during the pandemic. A global landscaping assessment from the Women's Refugee Commission shows adaptations to service provision across humanitarian settings included multi-month dispensing of supplies, task-shifting to include community-based service delivery, promoting self-administration of injectable contraceptives and other self-care methods, and integrating contraceptive service delivery with the provision of other essential health services [30]. Together with our study results, there is evidence that extending and institutionalizing these adaptations can help address pre-existing service gaps.

In addition to project-level adaptations, the literature shows that policies to support reproductive, maternal, newborn, child and adolescent health services were rapidly developed by countries in sub-Saharan Africa, further supporting the rollout of essential FP/RH services during COVID [41]. Project adaptations implemented by both projects were consistent with existing policies and guidance [42]. Policies focused on maintaining continuity of FP/RH services provision along with key infection prevention measures such as use of PPE, social distancing, triaging patients in facilities and reducing the number of people in facilities at once. All of these were described by project staff as key elements of their strategy to continue services. Additionally, projects were able to creatively find solutions and adapt to their specific challenges in ways that also were not specified by the policies. For example, they shifted to community-level outreach, batched clients, created appointment systems, placed handwashing stations in the community, and learned to make their own hand sanitizer.

Persistent challenges to the delivery of health care in limited resource and humanitarian settings such as adequate human resources, materials and supplies required for health care provision, and FP/RH health in particular were noted. Health care workers described needing better training, adequate remuneration, psychological support and opportunities for advancement. Delivering FP/RH services in humanitarian settings continues to be a challenge due to logistic and cultural factors and participants noted the need for increased education and sensitization—targeting both men and women, as well as further community outreach and engagement. This need for community-based education and services is also emphasized by Munyuzangabo et al. in their study on the delivery of FP/RH services in humanitarian settings [43], as well as by Lusambili et al. In their qualitative study on the impact of COVID-19 on RMNCH care among refugee women in Africa [37].

Finally, participants describe how COVID-19 brought long lasting benefits such as: improvements to organizational processes, the chance to learn new clinical skills and improve time management, along with increased awareness of the importance of adaptability, and strengthened interpersonal relationships and communication skills. Ensuring these benefits can be built upon is essential to capitalizing on the ingenuity and strength demonstrated by project staff.

## Limitations

As quantitative data came from physical project registers, it was difficult to obtain and verify data located within humanitarian settings experiencing conflict. UNHCR-SS-HNIR data was unavailable for family planning in both November and December and months with incomplete data could not be assessed as travel to sites was limited due to security issues and pandemic travel restrictions. Quantitative programmatic data was limited to 2020 and analyses were descriptive, so results do not account for any potentially confounding factors such as seasonality of service delivery or uptake. As more extensive historic data prior to the start of the COVID-19 pandemic were not available, we are limited in our ability to assess trends in FP service provision. The quantitative analysis is cross-sectional it is not possible to determine whether the adaptations described were the cause of the changes in programmatic outcomes. It should also be noted that we were not able to obtain the client perspective on the adequacy of services due to challenges in remote data collection during COVID-19; this is a vital perspective on accessibility and acceptability of health services that should be explored.

## Conclusions and implications for programs

COVID-19 required humanitarian projects to act beyond their usual, significant responsibility for providing food, housing, security and health care to affected populations. Qualitative and quantitative data show both that lockdowns had significant effect on provision of services and that protective measures and adaptations to service provision allowed the continued provision of essential FP/RH services without significant negative decline. Additionally, project staff reported that modifications that emerged during COVID-19 potentially have systematic and personal benefits lasting beyond COVID-19. Our data demonstrates that when protective measures are rapidly implemented, projects can demonstrate resilience in the face of a new threat disrupting both the health care and broader social systems. These findings are consistent with other studies showing that innovative adaptations to FP/RH service provision in humanitarian settings can help mitigate the impact of health emergencies. The authors hope that this qualitative documentation of adaptations and their perceived impact can lead to the creation of a "menu" of possible adaptations and specific policies for the provision of FP/RH care in humanitarian settings. Already, data from this study and the COVID-19 response of both

projects has been shared at virtual humanitarian coordination meetings held at the facility, local, and state levels, to note progress and challenges of program implementation within humanitarian settings. Within the IHANN II project, the methods utilized for addressing COVID-19 including early detection and management of disease have been adapted to inform the integrated community cases management (ICCM) of geriatric, feverish illness, and non-communicable disease cases. Additionally, despite these encouraging results, more resources and support must be provided to health care workers and project staff working under these challenging conditions—financial, human resources, psychological, and logistic support is essential.

Adapting to COVID-19 in humanitarian settings required effort, flexibility and creativity. A resilient health system is one that can prepare for and withstand the stress of shocks such as disease outbreaks, natural disasters and civil unrest in a way that protects both human life and the larger system [44]. Resilience is essential to achieving good outcomes before, during and after a disaster. A key element of resilient systems is adaptability, the ability to transform to improve function under adverse conditions in a way that enhances performance in the short and, ideally, the long term [44]. Given the challenges existing in humanitarian settings, this resilience is even more remarkable.

## Acknowledgments

We wish to acknowledge the leadership provided by the Federal and State Government of Nigeria and the heroic work of health care workers in health facilities, communities, and camps for internally displaced persons. We thank the staff of the IHANN II and UNHCR-SS-HNIR projects especially those who graciously shared their time and thoughts during interviews. Finally, we appreciate the insights provided by Robert Chiegil.

## Author Contributions

**Conceptualization:** Emily Evens, Bamidele Bello, Kate Murray, Nadia Tefouet, Adesegun Fatusi, Bridget Nwagbara, Mercy Riungu, Hadiza Khamofu, Jean Christophe Fotso, Ndola Prata.

**Formal analysis:** Emily Evens, Ashley Ambrose, Bamidele Bello, Kate Murray, Nadia Tefouet, Ndola Prata.

**Investigation:** Bamidele Bello, Kate Murray, Ndola Prata.

**Methodology:** Jean Christophe Fotso, Ndola Prata.

**Project administration:** Bridget Nwagbara, Mercy Riungu, Tijani Maji.

**Supervision:** Emily Evens, Bamidele Bello, Adesegun Fatusi, Jean Christophe Fotso.

**Validation:** Bridget Nwagbara, Mercy Riungu, Tijani Maji, Hadiza Khamofu.

**Visualization:** Ashley Ambrose.

**Writing – original draft:** Emily Evens, Ashley Ambrose, Jean Christophe Fotso, Ndola Prata.

**Writing – review & editing:** Emily Evens, Ashley Ambrose, Bamidele Bello, Kate Murray, Nadia Tefouet, Adesegun Fatusi, Bridget Nwagbara, Mercy Riungu, Tijani Maji, Hadiza Khamofu, Jean Christophe Fotso, Ndola Prata.

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
