## [Decision Letter · Decision Letter 0]

2 Mar 2023

PGPH-D-22-01594

“Africans, we know how to adapt indeed”: Adaptations to family planning and reproductive health services in humanitarian settings in Nigeria during the COVID-19 pandemic

Dear Dr. Ambrose,

Thank you for submitting your manuscript to PLOS Global Public Health. After careful consideration, we feel that it has merit but does not fully meet PLOS Global Public Health’s publication criteria as it currently stands. Therefore, we invite you to submit a revised version of the manuscript that addresses the points raised during the review process.

We look forward to receiving your revised manuscript.

Kind regards,

Hannah Tappis, DrPH, MPH

Academic Editor

Journal Requirements:

Reviewers' comments:

Reviewer's Responses to Questions

**Comments to the Author**

1. Does this manuscript meet PLOS Global Public Health’s publication criteria? Is the manuscript technically sound, and do the data support the conclusions? The manuscript must describe methodologically and ethically rigorous research with conclusions that are appropriately drawn based on the data presented.

Reviewer #1: Yes

Reviewer #2: Yes

2. Has the statistical analysis been performed appropriately and rigorously?

Reviewer #1: N/A

Reviewer #2: No

3. Have the authors made all data underlying the findings in their manuscript fully available (please refer to the Data Availability Statement at the start of the manuscript PDF file)?

Reviewer #1: Yes

Reviewer #2: Yes

4. Is the manuscript presented in an intelligible fashion and written in standard English?

Reviewer #1: Yes

Reviewer #2: No

5. Review Comments to the Author

Reviewer #1: It is good to see data on what happened to SRH service provision in humanitarian settings during Covid. I have a few minor comments – mostly questions of clarity.

• You mention lockdowns in some states. Can you provide any detail for what the Covid restrictions were in the 2 states where the study took place (Borno and Cross River)? For example, how long were lockdowns?

• Are all of the health facilities included in this study public health facilities? Or are some NGO-run?

• Table 1: I’m not sure why you have FP as well as Referrals for FP. It might be helpful to indicate which methods are available through direct service delivery vs via referral?

• P.6: why was OPV included in the facility datasets given the focus on FP/RH services?

• P.6: You say the facility data is January – Dec 2020, but then later say you compare to 4 months prior to March 2020 – wouldn’t that include Nov and Dec 2019?

• P. 6: Could you add a table that shows how many of each category of IDI respondent? Were all interviews conducted in English?

• P.8: I’m a little confused – according to the Categories you mention, FP services were paused during Covid (“Category 2 patients were advised to stay at home unless non-preventive care was needed.) But then, later you say that provision of FP services “showed little effect’ – does this mean they ignored the guidance or those guidelines were not in fact implemented? Related to this, you mention what the policies were, but perhaps could be a bit more clear on how those policies were implemented?

• P.9, last paragraph: you refer quite a bit to the triage and how that helped. Could you perhaps explain a bit more what this new triage was?

• P.14: “Some participants noted that ANC visits and facility-based delivery increased during

• COVID-19 because of the process of identifying people needing care in the community and referring them to the hospital.” Could you explain this statement a bit more? I’m not sure what “process” they refer to or how it changed?

• P. 17, last paragraph: You may want to cite supporting literature from the West African ebola epidemic that also showed FP visits rebounding after sharp decreases during the height of the epidemic.

• Given that in many other places SRH services were highly impacted by Covid-19, could you say a bit more about why this was different?

• Can you say anything about how these programs will use these data to address gaps?

Reviewer #2: This manuscript presents a case study of how two projects being implemented in a humanitarian setting were able to adapt to ensure continuity of services during the COVID-19 pandemic. The article makes a great contribution to our understanding of how programs adapted during the pandemic. However, there are several issues that need to be addressed to improve the paper.

Title

The title is not all-encompassing as it references FP/RH services, yet the article includes child immunization services.

Abstract

The structure of the abstract needs to be revised. Lessons learned ideally should come last. In the result section, it might be better to present the quantitative data first before the qualitative data.

Methods

• Provide some background information on the health facilities where data was collected.

• Provide the rationale for restricting the analysis to a 4-month pre-COVID period. Are the 4 months adequate to illustrate the trend in FP service provision accounting for seasonality differences?

Results

• Outline the 8 policies that were reviewed.

• Add more verbatim quotations to illustrate the qualitative findings. For instance, in the COVID-19 adaptations-no qualitative quotes have been presented.

• For ease of reviewing the graphs, indicate when COVID began, and when services were adapted.

• On the quantitative data, consider including the median(and IQR) number of attendances pre-COVID compared to during the COVID period and after.

• The qualitative results highlight the changes that were made without presenting what was happening before. For instance, IHANN staff worked two-week shifts, which was an adaptation but was happening prior to this change to help understand the changes that were made.

• On family planning, the authors indicate that disruptions caused some patients to change methods to what was available. Provide more information on the contraceptive methods that were stocked out during this period.

• In several instances, it is unclear who is referenced by the quotes. In many instances, the authors have indicated participants stated. Considering that the interviews were conducted among healthcare providers, project staff, and outreach workers from the two projects, it will be helpful if the authors could consistently indicate whose verbatim quotations are being referenced. Given the variability in the service uptake data between the two projects, was there an analysis of the qualitative data to help provide additional insights on what could have driven the differences considering all the study sites were from the humanitarian setting?

Limitation

The limitation section should come at the end of the discussion section.

Discussion

• Several statements have been made without being referenced. For instance, the first sentence-FP/RH is a key and often unfulfilled need of women and girls in humanitarian settings.

• Discussion section mostly paraphrases the results. There is little reference to existing literature to explain the findings.

• Explain variability in the results presented. For instance, in the IHANN service trends, there was a reduction in the uptake of ANC services between April to June, but that same pattern is not replicated among pregnant women receiving HIV testing and counseling. FP services also seem unaffected.

• Additionally, the IHANN trends also differ from UNCHCR service trends. Provide more insights on these differences considering that both data were from the humanitarian setting.

6. PLOS authors have the option to publish the peer review history of their article (what does this mean?). If published, this will include your full peer review and any attached files.

**Do you want your identity to be public for this peer review?** For information about this choice, including consent withdrawal, please see our Privacy Policy.

Reviewer #1: No

Reviewer #2: No

---

## [Editor Report · Decision Letter 1]

7 Jun 2023

“Africans, we know how to adapt indeed”: Adaptations to family planning and reproductive health services in humanitarian settings in Nigeria during the COVID-19 pandemic

PGPH-D-22-01594R1

Dear Ms. Ambrose,

We are pleased to inform you that your manuscript '“Africans, we know how to adapt indeed”: Adaptations to family planning and reproductive health services in humanitarian settings in Nigeria during the COVID-19 pandemic' has been provisionally accepted for publication in PLOS Global Public Health.

Best regards,

Hannah Tappis, DrPH, MPH

Academic Editor